# *Salmonella enterica* Serovar Gallinarum Biovars Pullorum and Gallinarum in Poultry: Review of Pathogenesis, Antibiotic Resistance, Diagnosis and Control in the Genomic Era

**DOI:** 10.3390/antibiotics13010023

**Published:** 2023-12-25

**Authors:** Mouad Farhat, Slimane Khayi, Jaouad Berrada, Mohamed Mouahid, Najia Ameur, Hosny El-Adawy, Siham Fellahi

**Affiliations:** 1Department of Veterinary Pathology and Public Health, Agronomy and Veterinary Institute Hassan II, BP 6202, Rabat 10000, Morocco; mouad.farhat96@gmail.com (M.F.); j.berrada@iav.ac.ma (J.B.); 2Biotechnology Research Unit, Regional Center of Agricultural Research of Rabat, National Institute of Agricultural Research, Avenue Ennasr, Rabat Principale, BP 415, Rabat 10090, Morocco; slimane.khayi@inra.ma; 3Mouahid’s Veterinary Clinic, Temara 12000, Morocco; mohamedmouahid@gmail.com; 4Department of Food Microbiology and Hygiene, National Institute of Hygiene. Av. Ibn Batouta, 27, BP 769, Rabat 10000, Morocco; najiaameur22@gmail.com; 5Institute of Bacterial Infections and Zoonoses, Friedrich-Loeffler-Institut, 07743 Jena, Germany; hosny.eladawy@fli.de; 6Faculty of Veterinary Medicine, Kafrelsheikh University, Kafr El-Sheikh 35516, Egypt

**Keywords:** *Salmonella* Gallinarum, *Salmonella* Pullorum, diagnosis, vaccination, virulence, multi-drug resistance

## Abstract

*Salmonella enterica* subsp. *enterica* serovar Gallinarum (*S*G) has two distinct biovars, Pullorum and Gallinarum. They are bacterial pathogens that exhibit host specificity for poultry and aquatic birds, causing severe systemic diseases known as fowl typhoid (FT) and Pullorum disease (PD), respectively. The virulence mechanisms of biovars Gallinarum and Pullorum are multifactorial, involving a variety of genes and pathways that contribute to their pathogenicity. In addition, these serovars have developed resistance to various antimicrobial agents, leading to the emergence of multidrug-resistant strains. Due to their economic and public health significance, rapid and accurate diagnosis is crucial for effective control and prevention of these diseases. Conventional methods, such as bacterial culture and serological tests, have been used for screening and diagnosis. However, molecular-based methods are becoming increasingly important due to their rapidity, high sensitivity, and specificity, opening new horizons for the development of innovative approaches to control FT and PD. The aim of this review is to highlight the current state of knowledge on biovars Gallinarum and Pullorum, emphasizing the importance of continued research into their pathogenesis, drug resistance and diagnosis to better understand and control these pathogens in poultry farms.

## 1. Introduction

The increasing demand for meat, driven by factors such as population growth, rising wage levels, and urbanization, has resulted in a significant rise of poultry meat production. Over the years, chicken’s share of global meat production has steadily increased, reaching 36% in 2016 from just 12% in 1961, with projections indicating its continued growth surpassing other meat production types by 2050 [1]. Concomitantly with the expansion of poultry production, comes the concern of food-borne diseases, with *Salmonella* being a leading cause of illness and mortality [2]. *Salmonella* is a rod-shaped, Gram-negative, facultative anaerobic bacterium belonging to the *Enterobacteriaceae* family. It encompasses two main species, *Salmonella bongori* and *Salmonella enterica* (*S. enterica*), within which more than 2000 different serovars capable of infecting humans and animals, including poultry, have been identified [3].

Among the various serovars of *S. enterica*, *Salmonella* Gallinarum biovars Gallinarum (*S*G) and Pullorum (*S*P) are notable for their unique characteristics. Unlike most *Salmonella* members, *S*G and *S*P are non-flagellated and non-motile. These biovars are associated with clinical illnesses in poultry and cause significant economic losses for farmers, particularly in developing countries, due to direct losses, flock replacement, and treatment expenses [4]. *Salmonella* infections caused by invasive serotypes, including *S*G and *S*P, can be lethal and require appropriate antibiotic treatment and control measures. However, the emergence of multi-drug resistant (MDR) strains of *Salmonella* has significantly impacted the effectiveness of antibiotic therapy, potentially leading to associated higher mortality rates [5]. To ensure effective control of Pullorum disease (PD) and fowl typhoid (FT), an organized national regulatory effective system is necessary for commercial poultry production [2].

Differentiation between *S*G and *S*P is crucial from an epidemiological and preventive standpoint [6]. Conventional diagnostic approaches for these *Salmonella* serovars require significant time and resources [7]. However, recent advances in molecular biology have opened up new possibilities for molecular-based diagnostic tests, enabling the identification of disease-causing agents at the species/subspecies/type level and monitoring the effectiveness of disease management programs [8]. In complement of several reviews published on *S*G and *S*P in the international literature [6,9], this review aims at exploring the recent developments and advances in the diagnosis and control of *S*G and *S*P.

## 2. *Salmonella* Gallinarum/Pullorum’s Genome and Its Relationship with Virulence

The genomes of *S*G and *S*. Enteritidis (*S*E) show a close relationship, which suggests the latter is a straight evolutionary descendent of the former, with *Salmonella* Gallinarum having a significantly higher number of predicted pseudogenes [10].

*S*G and *S*P, like other typhoid serovars, affect birds by the oral route, gaining intestinal epithelial or lymphoid tissue cells in the Peyer’s patch and caecal tonsils. Free bacteria, alongside infected phagocytes, migrate to lymphoid tissues where they multiply. They return to lymphoid tissue in the intestine through an entirely unidentified process, where they are shed in feces [6].

In the disease course, the progression of *S*G depends on the bacteria’s capacity to survive and multiply inside the liver and spleen’s macrophages. Several virulence genes are activated by *S*G in order to withstand phagocytosis and bacterial clearance, infect the host, and multiply [11]. In the early stage of infection, and being devoid of flagella, *S*P and *S*G have the advantage of entering the digestive tract via Toll-Like Receptor 5 without inducing a significant inflammatory reaction, promoting systemic infection, and perhaps exhibiting a special avian host adaptation [12]. This is caused by mutations in genes responsible for the synthesis of the flagellar structure (*flh*A, *flh*B, *che*M, *flg*K, and *flg*I), even if the *fli*C gene is intact [13]. When it comes to intestinal colonization, different fimbrial operons can be distinguished. Of 13 fimbrial operons detected by Thomson et al. (2008), only *fim*, *bcf*, *cSG*, and *ste* remain undisrupted, although std is missing in *S*G. However, the other eight fimbrial operons (*sef*, *peg*, *lpf*, saf, *stf*, *stb*, *sth*, and *sti*) are mutated. Meanwhile, three genes generating fimbrial proteins replace the five *pef* operon genes that are present in the *S*E virulence plasmid [10].

To cause intestinal inflammation, different TTSS (Type Three Secretion System) effector proteins genes are required, such as sopE cassette gene [14]. Meanwhile, some genes responsible for enteritis like *sop*A, *pip*B, *sif*B, and *big*A are truncated, which may affect the entero-pathogenicity of this serovar [15]. Several potential virulence genes distinct from those found in *S*E and *S*G have been discovered in *S*P using suppression subtractive hybridization [16]. These include *Ipaj*, which is a gene for Colicin Y production, the *fae*H and *fae*I fimbrial plasmid genes, and the *tra*G, which produces a coupling protein with the *Tra*J gene, playing a role in DNA translocation [16].

In the intracellular level, *S*G has lost many metabolic pathways, notably the inaptitude to use 1,2-propanediol, due to mutations in the *ttr*, *cbi*, and *pdu* operons responsible for tetrathionate respiration, coenzyme B12 biosynthesis (B12; *cobalamine*), and 1,2-propanediol degradation, respectively [17]. *S*G is also unable to assure the decarboxylation of ornithine due to mutations in the gene *spe*C, as opposed to *S*P which can metabolize it [18]. Furthermore, the glycogen metabolism is altered through mutations in the genes responsible for glycogen production: *glg*A, *glg*B, and *glg*C. Concerning *glg*C, it was suggested that deletions in *S*P and *S*G are not the same [19]. It is believed that the intracellular multiplication of *S*G is maintained by homologous/orthologous genes, which increases virulence in *S*. Typhimurium, mainly SPI-2 (*Salmonella* Pathogenicity Island-2) genes [20,21]. In *S*P, it is thought that SPI-2 contributes to persistent infection, probably in the initial survival within macrophages [22]. In addition, *sly*A, a key regulatory component in SPI-2 expression, was found in all of the 94 *S*G strains studied by Agrawal et al. (2005) [23].

*Salmonella* Gallinarum also lacks the genes that allow the use of the substitute electron acceptor dimethyl sulphoxide (dmsA1, dmsA2) and trimethylamine N-oxide (*tor*S), along with other genes related to colonization like the SP-I3 *shd*A and *shd*B genes. The latter are also proven to code for efficient and prolonged fecal shedding [10]. Additionally, *bcSG* is a primordial gene for the survival of the bacteria outside the host, and in *S*E, it contributes to the production of biofilm. In *S*G, this gene is mutated, which explains its inability to produce cellulose, and thus its poorer survival outside the host [10].

## 3. Multi-Drug Resistance: A Worldwide Threat

In several countries, PD and FT are still being treated with antimicrobial therapy. While several chemotherapeutic drugs have proven to be effective in lowering mortality rates, they cannot completely eradicate infection within a group of birds [24]. This widespread use of antibiotics in poultry farming, and their use as growth promoters has favored the emergence of resistant bacterial species that can spread to humans through the food chain, aggravating the problem of antimicrobial resistance worldwide [25].

There are two types of antibiotic resistance: innate and intrinsic. The intrinsic mechanism of resistance includes the following mechanisms: change of the antibiotic target site, permeability of cell membranes, efflux pumps that carry antibiotic molecules out of the cell, and antibiotic inactivation [26]. Conversely, acquired resistance results from bacteria appropriating genetic material through horizontal gene transfer. Bacteria can acquire genes that confer antibiotic resistance to develop an antibiotic-resistant phenotype. These genes are carried and transferred by integrons, transposons, plasmids, and prophages [27,28]. Different resistance mechanisms are deployed by *Salmonella* depending on the antibiotic family in question [28]. For instance, resistance against aminoglycosides, which acts by binding on the ribosome, can be provided by the following: (i) methylation of the subunit 30S of the ribosome, (ii) aminoglycoside acetyltransferase coded by aac(6′)-Ib, and (iii) decreased permeability [29,30]. Resistance against β-lactams, which acts by interfering with the synthesis of peptidoglycan, is provided by the following: (i) enzymatic inactivation through the hydrolysis of β-lactamase, (ii) expression of the β-lactams resistance gene bla, (iii) β-lactamase point mutation, which produces an extend-ed-spectrum β-lactamase, (iv) overexpression of the efflux pump genes *mac*AB, *mdt*ABC, *emr*AB, *mdt*K, and *acr*D, and (v) diminished permeability [31]. Quinolones act by interfering with bacterial DNA replication and transcription, and resistance against them is provided by the following: (i) expression of the quinolone resistance genes *par*C and *gyr*A, (ii) reduced activity by the aac(6′)-lb-cr gene expression, and (iii) efflux pumps encoded by the *oqx*AB and *qep*A genes [32]. Macrolides and chloramphenicol bind to the 50S subunit of ribosomes and inhibit the production of proteins. Mechanisms responsible for the resistance against macrolides are the following: (i) enzymatic inactivation by phosphotransferase or esterase, (ii) mutations in the 23S rRNA gene, and (iii) efflux pumps genes *mef* and *msr* [33,34]. Resistance against chloramphenicol is provided by the following: (i) expression of the *flo*R resistance gene, (ii) enzymatic inactivation, and (iii) overexpression of acrAB-tolC efflux system [34,35]. Resistance mechanisms against tetracyclines (TET), which bind to the 30S ribo-somal subunit and inhibit protein synthesis, are as follows: (i) target site mutation in ribosomes, (ii) expression of the genes *tet*A, *tet*G, and *tet*B, and (iii) efflux pump against tetracyclines [35]. Sulfonamides prevent bacteria from producing the B vitamin folate, and the resistance mechanism against them is the expression of *sul*1, *sul*2, and *sul*3 genes encoding dihydropteroate synthetase (DHPS) with low affinity for sulfonamides [36].

Several studies have addressed the emergence of *S*G and *S*P antibiotic resistance. In Africa, a review focusing on poultry farming systems, antimicrobial use (AMU), antimicrobial resistance (AMR), and circulating serotypes of *Salmonella* from January 2010 to December 2020 was published by Ramtahal et al. (2022). A total of 122 studies were subjected to this review, and 11.5% of the studies included descriptions of AMU, which differed between and within countries. Thirty investigations found the presence of multidrug-resistant (MDR) in *Salmonella* isolates, with a prevalence ranging from 12.1% in Zimbabwe to 100% in Senegal, Nigeria, Ethiopia, South Africa, and Egypt [37].

Seo et al. (2019) conducted a study examining AMR in *S*G isolates from 2014 to 2018 in South Korea. A total of 130 *S*G isolates were collected from poultry farms with fowl typhoid outbreaks. These isolates showed resistance rates at 78.5%, 52.5%%, 26.9%, and 14.6% to nalidixic acid (NAL), gentamicin (GEN), ciprofloxacin (CIP), and ampicillin (AMP), respectively. The amplification of resistance genes showed that: out of 36 GEN-resistant isolates, 61.1% carried the ant(2”)-I gene; 52% of the 25 β-lactam-resistant isolates carried the blaTEM-1 gene; out of 13 trimethoprim-sulfamethoxazole-resistant strains, 69.2% and 23.1% harbored sul1 and sul2 genes, respectively, with 15.3% harboring both genes; of 7 chloramphenicol(CHL)-resistant strains, 42.8% carried the *cmlA* gene; and in 7.7% of the 39 NAL-resistant strains, the *qnrB* gene was detected. Furthermore, there were significant increases in the occurrence of resistance to the following antibiotics: AMP, amoxicillin-clavulanic acid (AUG2), NAL, CIP, chloramphenicol (CHL), and colistin (COL). Alarmingly, the number of MDR isolates increased rapidly from 23.1% in 2014 to 60.7% in 2018 (*p* < 0.05) [38].

Another study conducted by Zhang et al. (2022) focused on the antimicrobial resistance and genotypes of *S*G isolates. The isolates exhibiting a multidrug-resistant phenotype showed resistance to at least 3 out of 18 tested antimicrobials. The most prevalent resistance profile was against streptomycin, sulfisoxazole, colistin, nalidixic acid, ciprofloxacin and gentamicin. This study also highlighted the horizontal contamination and spread of multidrug-resistant strains inside and between different companies [39].

Additionally, the prevalence and genetic content of class 1 integrons in 90 *S*G isolates between 1992 and 2001 were investigated by Kwon et al. (2002). Out of the examined strains, 39% carried class 1 integrons and three different sizes of amplicons containing resistance cassettes were identified. These integrons conferred resistance against aminoglycosides (*aad*A1a, *aad*A1b, *aad*B and *aad*A2) and trimethoprim (*dhfr*XII). The study revealed that the prevalence of integron-carrying strains of *S*G increased over time and acquired additional resistance cassettes. Thus, the presence of integrons poses a potential threat to the effectiveness of antibiotic treatment in FT [40]. This was supported by Gong et al. (2013), who conducted a study highlighting the association between class 1 integrons and MDR in *S*P in eastern China, where all class 1 integron-positive isolates displayed MDR and demonstrated higher resistance levels than integron-negative isolates. It also showed that the prevalence of MDR strains was low (9.4%) from 1962 to 1968 but increased significantly between 1970 to 1979 and from 1980 to 1987 (64.6 to 78.7%), reaching 96.6% between 1990 and 2010 [41].

Enrofloxacin was used routinely in Korea to treat bacterial infections in poultry until it was prohibited for use in industrial layers in 2017 [38]. Since enrofloxacin is metabolized to ciprofloxacin, the resistance to both is impacted by the use of enrofloxacin [38,42]. According to Lee et al. (2004), there has been a rise in resistance to fluoroquinolones among Korean *S*G strains, with 6.5% and 82.6% increases for enrofloxacin and ofloxacin, respectively. This resistance is due to a mutation in the *gyr*A gene. On the other hand, these strains have also developed resistance to other antibiotics, including ampicillin (with a resistance rate of 13%), gentamicin (43%), and kanamycin (69.6%) [43]. Furthermore, a study was conducted to analyze alterations in antibiotic resistance patterns of *S*P strains that were collected from chickens in China from the years 1962 to 2007. The study discovered that ampicillin, streptomycin, tetracycline (TET), carbenicillin, trimethoprim, and sulphafurazole had high levels of resistance. Moreover, it was observed that there was an increase in the number of multi-resistant strains between 2000 and 2007, implying that a more rational use of antibiotics is needed [44].

Antibiotic resistance was also the subject of a study by Penha Filho et al. (2016), where the authors compared the susceptibility of 8 *S*G and 1 *S*P from the period 1987–1991 to 24 *S*G and 17 *S*P from the period 2006–2013. The results showed that from 1987 to 1991, all *S*G and *S*P were susceptible to the 14 antibiotics tested, namely:AUG2, cefotaxime, ceftazidime, cefepime, aztreonam, ertapenem, ceftiofur, enrofloxacin, tetracycline, chloramphenicol, florenfenicol, trimethoprim-sulfamethoxazole, ciprofloxacin, and nalidixic acid. However, from 2006 to 2013, the *S*G strains’ susceptibility to most of the non-β-lactams decreased from 100% to the following: nalidixic acid (58%), ciprofloxacin (63%), enrofloxacin (67%), tetracycline (92%), florenfenicol (96%), and trimethoprim-sulfamethoxazole (96%). For *S*P strains, it decreased to the following: nalidixic acid (65%), ciprofloxacin (71%), enrofloxacin (94%), and tetracycline (94%) [45].

More recently, in a study by Farahani et al. (2023), antibiotic resistance of 60 *S*G isolates was examined and the rate among isolated strains was as follows: 100% for penicillin, 80% for nitrofurantoin, 75% for amoxicillin, 50% for amoxicillin-clavulanic acid, 45% for nalidixic acid, 30% for neomycin sulfate, 20% for chloramphenicol, and 5% for ciprofloxacin. On the other hand, colistin, kanamycin, imipenem, ertapenem, ceftriaxone, ceftazidime, and trimethoprim-sulfamethoxazole were all effective against all isolates. Additionally, the expression of the resistance genes IMP, VIM, NDM, and DHA, coding for beta-lactamases, and the gene *qnr*A, coding for quinolone resistance, was absent in all 60 isolates. In contrast, the expressed resistance genes are GES (85%), *bla*_OXA48_ (60%), SHV (60%), CITM (20%), FOX (10%), Fox M (70%), KPC (15%), MOXM (5%), coding for β-lactamases, and *qnr*B (75%) and *qnr*S (5%), coding for quinolones resistance [46]. This study also highlighted the correlation between antibiotic resistance and biofilm forming.

The latter is a set of an extracellular matrices and persistent cells that we can find both inside and outside the host body [47]. The capacity of bacteria to form biofilms affords a favorable exchanging space where the frequency of genetic material exchange is more important. Thus, genes encoding resistance are transferred among bio-film-forming bacteria with MDR traits [48,49]. This was confirmed by the results of the study by Farahani et al. (2023), where the findings demonstrated a positive correlation between the degree of biofilm formation and certain resistance genes, including Fox M, GES, Fox, KPC, and *qnr*B [46].

These findings, summarized in Table 1, collectively underscore the growing challenge of multi-drug resistance in *S*G and *S*P and emphasize the urgent need for effective strategies to control and manage these pathogens [39,50].

## 4. Diagnostic Advances in *Salmonella* Gallinarum/Pullorum

In the last three decades, there has been significant interest in *Salmonella* organisms because of their pathologic occurrence in humans and animals, as well as their ease of culture and genetic manipulation. Technological progress led to a biological revolution in terms of genetic and immunological information on *Salmonella*, which lowered the bar for the challenge of finding new approaches to control this disease [6].

Genotypic techniques accessing genetic material from chromosomal and extrachromosomal DNA allow for differentiation between closely related strains. These include multiple polymerase chain reaction (PCR), Pulsed-Field Gel Electrophoresis (PFGE), DNA microarray, and sequence-based techniques [51]. In epidemiological research, these techniques have been combined with well-established standard approaches, including serotyping and phage typing, to differentiate and map strains more precisely [52].

### 4.1. Polymerase Chain Reaction

In epidemiological research of *S*G and *S*P, PCR is in great demand thanks to its rapidity and accuracy, whether for detecting *S*G and *S*P or for differentiating between them. Thus, a large spectrum of genes is to be examined [53].

For instance, a one-step multiplex PCR assay was developed by Zhu et al. (2015) to detect the most common serovars of *S. enterica* subsp. *enterica* in chickens. This technique makes use of primers that amplify various sets of DNA sequences linked to various *Salmonella* species and serovars in a single PCR tube. They used a 4-nucleotide deletion in the *ste*B gene of the *S*P biovar to create primers for the *ste* gene that are unique to the *S*G serovar. All strains investigated, except for *S*P, produced amplicons in response to these primers; nevertheless, the researchers found that primers targeting the rhs gene were different for *S*G and *S*P [54].

In 2016, Xiong et al. developed a PCR primer based on the *flh*B gene, which encodes the flhB membrane protein, a component of the flagellar secretion system. According to the findings, *S*P and *S*G had a deletion in this gene compared to other serovars [55]. A year later, to distinguish *S*P and *S*G from other serovars, Xiong et al. expanded on their earlier work by using the *flh*B gene as a PCR target and adding primers for the *tcp*S and *lyg*D genes to create a multiplex PCR that could distinguish between *Salmonella* Dublin, *S*E, and *S*G/*S*P. Previous genomic data, revealing that the *tcp*S gene is present in all serovars, the *lyg*D gene is only present in *S*E, and the *flh*B gene is present in *S*P/*S*G in a truncated form, guided the selection of primer targets. This allows for the precise identification of the targeted serovars, with a detection limit of no less than 100 CFU [56].

The genomic analysis of *Salmonella* serovars has also led to the discovery of other genes, such as the *cig*R gene found on SPI-3 which encodes c*ig*R, a putative inner membrane protein that does not affect the interaction with the host [57,58]. This gene was targeted in a study conducted by Zhou et al. (2020) to develop a quick, one-step multiplex PCR technique for the multiplex PCR system to detect *Salmonella* and precisely identify *S*G and *S*P. This gene was chosen because of a little change in sequence. Compared to other *Salmonella* serotypes, a segment of 42 bp is missing in *S*G and *S*P. This was shown by PCR, where the amplification products for *S*G and *S*P showed just one band (421 bp), in contrast with other *Salmonella* strains showing two bands (463 bp and 65 bp) [53].

Trying to differentiate between *S*G and *S*P, and to distinguish them from other well-known *Salmonella* serotypes belonging to serogroup D, Shah et al., (2005) developed an interesting allele-specific PCR method based on an *S*P/*S*G-specific nucleotide polymorphism discovered in the *rfb*S gene after examination using sequencing. Since sequencing showed that *S*G had guanine and adenine residues at positions 237 and 598, respectively, which are inverted in *S*P, these researchers developed primers specific for each serovar (*rfbS*G for *S*G and *rfbSP* for *S*P). PCR based on *rfbS*G primers demonstrated a distinctive 187 bp amplicon with the DNA of the *S*G strain but not with the other *Salmonella* strains of serogroup D, while PCR based on *rfbSP* primers generated a 187 amplicon in *S*P, *S*E, *S*. Dublin, and *S*. Typhi. In terms of sensitivity, this method was able to identify *S*G DNA at a concentration as low as 100 pg/L with 100% specificity [59]. Moreover, primer sets targeting the nucleotide polymorphism at position 237 were used in the investigation by Desai et al. (2005). In less than 3 h, these primers generated a 147 bp amplicon in *S*P alone, while *S*G and other *Salmonella* serotypes failed to yield any amplicons. As a result, it can be concluded that the primers utilized in this investigation are highly specific to *S*P. Additionally, this test was able to identify *S*P DNA at a concentration as low as 100 pg/l, demonstrating the high sensitivity of the allele-specific PCR test [60].

In 2011, Kang et al. developed a duplex PCR that specifically targets the *spe*C gene, (shared by both *S*P and *S*G biovars) and the *glg*C gene, which has an 11 bp deletion in *S*G. The primers used produced a 174 bp amplicon specific to *spe*C in *S*P and two amplicons of 174 bp and 252 bp specific to *spe*C and *glg*C, respectively, in *S*G. Therefore, the test accurately identified 53 isolates as *S*G and 21 isolates as *S*P out of a total of 131 strains of *Salmonella* and other related Gram-negative bacteria [61].

Xiong et al. (2018) successfully and concurrently identified and differentiated *S*P and *S*G by focusing on three genes: *stn*, *“I137 08605”,* and *rat*A. The “*I137 08605*” gene was found through bioinformatic analysis to be unique to *S*P and *S*G. Furthermore, the *ratA* of *S*P, a 4776 bp long gene, is 85% longer than the *rat*A gene of *S*G and other serovars. These researchers demonstrated that the *Salmonella* enterotoxin gene, *stn*, was unique to *S*E. The 290 bp amplicon specific to “*I137 08605*” for *S*P was also obtained using the primer sets employed in this multiplex PCR experiment, as were two other amplicons of 290 and 571 bp for *S*G specific to “*I137 08605*” and *ratA* ROD, respectively. According to these authors, DNA may be amplified at a concentration as low as 67.4 pg/L, indicating that this method has a high sensitivity and can be an effective diagnostic tool in veterinary clinical laboratories and epidemiological research [62]. Additionally, Xu et al. (2018) developed a quick and accurate PCR approach which targets the *ipaJ* gene for easier detection of *S*P. In a 2 h reaction, the PCR amplified a 741-bp product specific to *S*P. Additionally, no cross-reactivity was observed with genomic DNA from strains other than *S*P [63].

Thus, the amplification of these allele-specific methods for detecting *S*G and *S*P is specific, repeatable, and quicker than the traditional bacteriological methods, and could be a useful molecular tool for rapid, accurate diagnosis of FT and PD, opening the door for Next Generation Sequencing (NGS) for further analyses of different genes’ sequences [59,63] (Table 2).

### 4.2. Next Generation Sequencing

Nucleic acid sequencing enables distinction between strains that are closely related to a monogenic resolution [64]. In several investigations, this method has been used for the entire genome or for a designated section of the genome to differentiate between different foodborne pathogens [65]. *Salmonella* in poultry has also been studied with WGS (Whole Genome Sequencing); for instance, a comparative genomics analysis enabled the evaluation of the genotypic differences between *S*G and *S*E, revealing a pan-genome that is open and contains several virulence determinants, genomic islands, and antibiotic resistance genes. This would allow for a rapid and accurate diagnosis, a better identification and characterization of *Salmonella* strains, and the development of new vaccines for the effective control of these infections [66].

Genetic understanding of *S*G and *S*P and how it pertains to their pathogenicity is still developing [67]. In a study conducted by Rakov et al. (2019), 500 *Salmonella enterica* subsp. *enterica* genomes were examined to determine the allele distribution of virulence determinants. In addition to the *STM3031* and *Sse*K1 genes’ presence or absence, hierarchical clustering resulted in separating between *S*G and *S*P by alleles of 23 VFs (Virulence Factors) (*CIg*R, *Sop*D, *Sip*A, *Avr*A, *Pip*B2, *pag*M, *Ste*C, *Zir*S, *Srf*A, *Sif*A, *Sif*B, *Ste*A, *Omp*D, *Sse*B, *Sse*C, *Pag*D, *Sop*B, *Pip*B, *Gtg*A, *Sop*D2, *Omp*X, *Fim*H, and *Stf*H). *Salmonella* Pullorum was also separated into two clusters, with one cluster lacking *Srf*A and the other possessing distinct alleles for 11 VFs (*Sip*C, *Spi*C, *Sif*B, *Sop*D, *Sse*B, *Sse*C, *Sse*G, *stf*H, *Ais*, *Pip*B2, and *Bcf*D). The serovar’s Cluster 2 and serovar Gallinarum shared numerous alleles (*Bcf*D, *Spi*C, *Sse*G, and *Ais*) [68].

Langridge et al. (2015) analyzed the *S*G and *S*P genomes and discovered 231 and 212 conserved pseudogenes, respectively. Such high numbers are in line with an evolution that relies on the accumulating mutations eventually causing a gradual loss of the ability to code for proteins. It has been demonstrated that some of the genes, such as *cbi*, *pdu*, and *ttr*, playing a key role in intestinal colonization, harbor harmful mutations in the *S*G and *S*P genomes supposedly because of these organisms’ lack of a primary focus on the intestines in their infections [69]. A modest number of conserved, serovar-specific pseudogenes were discovered through genomic comparisons between various *S*G and *S*P genomes. Some of them were tested and validated with PCR [70].

Many studies have been published to distinguish between *S*G and *S*P as different biovars. For instance, Kwon et al. (2001) sequenced the hyper-variable region of the RNA polymerase beta subunit of the *rpoB* gene using the automated Sanger method. They discovered a 98% resemblance between *S*E and *S*G/*S*P serovars, while there was a 100% similarity between *S*G and *S*P. Additionally, *S*E and *S*G serovars have slight alterations (Arginine at position 247 by histidine and aspartic acid at position 254 by glycine, respectively). According to amino acid sequence analysis using serotype Typhimurium as a reference, *S*P biovar showed a similar pattern, (substitution of aspartic acid at position 281 by tyrosine). Consequently, this region may be a viable molecular marker to distinguish between *S*G and *S*P biovars [71].

Moreover, Feng et al., (2013) sequenced the whole genomes of *S*G and *S*P and found that they had few small differences. According to sequencing data, 14 genes, including the c-type cytochrome *tor*C and their two-component regulatory system *tor*R/*tor*T, were present in *S*G but absent in *S*P. Meanwhile, other genes, such as *mdt*I and *mdt*J, which are thought to be crucial mutation-coding proteins, were present in *S*P but not in *S*G [72].

Furthermore, *S*G has mutations in the *glg*A, *glg*B, and *glg*C genes, which are crucial to producing glycogen, while *S*P does not contain the same deletion in *glg*C. In fact, *S*G contains an 11 bp deletion (GATCGATCACG) which is absent in *S*P. Additionally, *spe*C mutations preventing *S*G from decarboxylating ornithine were found [10,61].

As already mentioned, both biovars exhibit substantial genome degradation that results in the production of pseudogenes. Each does, however, have a few special pseudogenes. For instance, *S*P alone possesses the virulence gene *sif*A, the DNA repair gene *mut*L, and genes encoding for enzymes involved in amino acid synthesis (*ilv*G, *ilv*I, and *trp*E) [73]. Of the 13 fimbrial operons inherited from *S*E, *Salmonella* Pullorum has just *saf*, *cSG*, and std intact, while *S*G lacks only the fimbrial operon std [56,58]. On the other hand, the three non-*fimbrial* adhesin genes (*shd*A, *sin*H, and *rat*B) in the CS54 island of the *fimbrial fae* operon are intact in *S*P but inactivated in *SG* [74].

Consequently, these methods would make it easier to incorporate recently obtained *Salmonella* WGS data into expanding pan-genome datasets. As a result, the reliability of epidemiological markers for outbreak analysis would increase. Moreover, setting up pan-genome baselines for different serovars would simplify the process of making interlaboratory comparisons, especially when dealing with extensive epidemics [75].

## 5. Vaccination against FT and PD

Vaccination is a common and time-tested method of protection against bacterial illnesses of relevance to animal health, including some serotypes of *Salmonellae* [76]. In the case of *S*P and *S*G, both live and killed vaccines have been developed to control these pathogens [77].

Killed vaccines are inactivated microorganisms that use various adjuvants to boost the immunogenicity. They have been used to safeguard poultry and their offspring against field threats. They enhance the circulation of antibodies, hence decreasing *Salmonella* shedding in the environment and its excretion in feces [78]. Nonetheless, they lack the cell-mediated immune response required to attack *Salmonella* because they express a reduced number of antigens and do not generate SIgA responses at mucosal surfaces [79,80].

On the other hand, live attenuated vaccines activate both cell-mediated and humoral immunity as well as the expression of all necessary antigens in-vivo, providing greater protection against *Salmonella*, especially by preventing adhesion, the first step to its colonization [81]. Live attenuated vaccine against FT was first used in 1956 by Herbert Williams Smith, when *S*G9R was developed. The lipopolysaccharide (LPS) structure of *S*G9R has a semi-rough texture that decreases the pathogenicity of this strain [82,83]. This vaccine also provides some degree of protection against *Salmonella* Enteritidis and *Salmonella* Typhimurium [84]. Even when given to young layer hens at the age of 4 weeks, the *S*G9R vaccination showed adequate safety and efficacy [85]. However, the *S*G9R vaccine strain has pros and cons; even if it assures good protection, its fecal shedding has been shown for up to 24 h following immunization and it may revert to the pathogenic smooth strain responsible for certain outbreaks [86].

Later, numerous vaccines using *S*G strains for FT prevention were tested, but only the *S*G9R strain is currently available for commercial use [87]. Other live vaccines have recently been generated utilizing genetic methods that delete many genes from the bacterial chromosome and reduce the likelihood of virulence returning [88].

In the 1990s, two experimental vaccines were investigated and shown to provide inadequate protection and environmental durability. The first one is an *aroAS*G mutant that was shown to be ineffective in safeguarding hens. It was taken orally and intramuscularly, and it remained in diverse tissues for a maximum of nine days. However, its level of protection in birds was less than the *S*G9R strain’s [89]. The second vaccine, *nuo*G, was developed by introducing a mutation into a highly pathogenic strain of *S*G. The induced protection was equivalent to or greater than *S*G9R and was less invasive in the gastrointestinal tract, liver, and spleen. However, the vaccine strain persisted for six weeks in the liver and spleen [90].

In 2000, the first report of lysogenization-related attenuation was made when a mutant of a P22 *sie* lysogenic wild *S*G strain was generated. In immunological protection studies, intramuscular injection of this strain yielded remarkable effects, providing one hundred percent protection against the homologous strain [91,92]. Additionally, a mutant with three deletions was examined as a potential live vaccine and yielded a protection comparable to the *S*G9R vaccine. However, liver and spleen bacterial counts were higher [92]. Researchers also generated a *met*C mutant of *S*G in 2007, and conducted several tests to assess how this mutation impacts virulence. Based on the results, a genetically engineered vaccine against FT using the *met*C mutant might be feasible in the future considering that this gene plays a key role in the virulence of *S*G in chickens [93]. In Brazil, researchers have created and investigated a mutant strain of *S*G that lacks the *cob*S and *cbi*A genes, both of which are required for the biosynthesis of cobalamin. This mutant strain showed effectiveness against *S*G wild-type-induced mortality in brown hens when administered once, with a systemic response, indicating that vaccinated birds develop Th1 and Th2 responses. In contrast, only vaccination with two doses demonstrated their efficacy in white birds. Thus, this *S*G mutant is a viable candidate that has shown good efficacy in preventing *S*G infection [11,94]. Another vaccine was developed by Matsuda et al. (2011) with a deletion of the *lon* and *cpxR* genes, which are engaged in bacterial invasion and multiplication within the host. The results showed a marked reduction in organ damage and a faster recovery from the strain challenge, offering safer and more effective protection than the *S*G9R strain [95].

In 2014, the intramuscular administration of a live attenuated *S*G vaccine, named JOL1355, was found to be safe in vaccinated hens, showing no side effects. Moreover, the vaccine demonstrated bacterial presence in the spleen for seven days following injection and only minimal visible lesions for up to three days after inoculation [96]. After 14 days of inoculation, the vaccine strain significantly contributed to the elimination of the wild-type *S*G strain used in the challenge from the internal organs. Therefore, this vaccination may be a great tool for generating acquired immunity and eliminating germs from diseased birds [96]. Another *S*G strain was studied in 2014 to investigate the crucial role of polyamines in the pathogenicity of *S*G. Data revealed novel approaches for producing inhibitors for these enzymes in FT therapies [78,97]. Later, mortality and clinical symptoms of a live attenuated *S*G *spi*C and *crp* deletion mutant were studied by Cheng et al. (2016) and it was determined that the mutant provided effective protection against FT [98].

Recently, in 2022, Senevirathne et al. developed an attenuated vaccine by removing the virulence-related genes *lon*, *cpx*R, and *rfa*L and reducing endotoxicity by removing the *pag*L open reading frame and substituting it with the *lpx*E gene from *Francisella tularensis*. These manipulations conferred to the mutant strain a detoxified lipid A structure, which induces a reduced inflammatory TNF-responses compared to the *S*G9R-based vaccine while significantly increasing IFN- cytokine levels, an adaptable marker of an antimicrobial reaction. As with the *S*G9R, subcutaneous immunization with this vaccine caused humoral and cell-mediated immune responses with Th1-skewed patterns, conferring disease protection. These results suggest that this detoxified *S*G strain decreased endotoxicity without affecting its protective effectiveness compared to *S*G9R [99].

For PD, vaccines did not appear until the early 21st century [87]. In order to identify the genes required for *S*P survival, Geng et al. (2014) generated and tested a signature-tagged mutagenesis (STM) bank of 1800 mutant hens. They further characterized one mutant, *spiC*, as a potential vaccine candidate. The mutant strain (spiCkm) was less virulent and more immunogenic than the parental strain. Additionally, no clinical symptoms were observed for four weeks, and this mutant was no longer isolated from organs eight days after infection, suggesting that this *spi*C mutant may be an effective option for a vaccination to prevent PD in chickens [100]. Based on these findings, Kang et al. (2022) developed a new vaccine (S06004ΔspiCΔrfaH) by deleting the *rfa*H gene, making it an LPS rough mutant. Thus, the O9 mono antibody did not agglutinate the mutant rough LPS phenotype strains. Additionally, the mutant strain showed a reduced bacterial colonization in the spleen and liver. Since the IgG titers against *S.* Pullorum were so high, it was clear that the immunized group had generated an excellent humoral immune response, and the analysis of lymphocyte proliferation and cytokine expression in the spleen also provided insight into the cellular immune reactions [101]. The s*pi*C gene was also targeted in a study by Wang et al. (2021) investigating the effect of a live attenuated vaccine with a mutated s*pi*C. The results showed increased expression of mRNA for the Th1 cytokines IFN- and IL-2 in the early stages and the Th2 cytokines IL-4 and IL-10 in the later stages, as well as an important humoral response confirmed by IgG and mucosal IgA titers. Cellular immunity was also stimulated by higher counts of CD3+CD8+ T cells and antigen-specific lymphocyte proliferation. Thus, this vaccine offered at least 90% immune protection when challenged with a wild *S*P strain and cross-protection to varying degrees against various *Salmonella* serovars, suggesting it as a safe and effective vaccine candidate [102].

In 2015, researchers studied the efficacy of an SPI-2 (*Salmonella* pathogenicity island 2) mutant of *S*P (S06004ΔSPI2) and discovered no indication of clinical complaints or changes in body weight. It was impossible to identify the mutant “S06004ΔSPI2” from the liver more than two weeks or three weeks after vaccination. Nevertheless, organs from the parent strain-infected group remained positive for three weeks following infection. Intriguingly, when chickens vaccinated orally with the *S*P mutant strain were challenged intramuscularly 10 days later with the *S*P parent, they had a 100% survival rate, but when challenged with phosphate-buffered saline, the vaccinated group only had about 60%. Meanwhile, in the case of *S*G challenge, the vaccinated group resulted in a 100% survival rate, while with PBS vaccination and challenge with the same strain resulted in a 30% survival rate. These findings suggest that the “S06004ΔSPI2” strain could be used as a live attenuated oral vaccine against both FT and PD [103]. Another vaccine against PD was developed by Guo et al. (2016) using an attenuated strain and protein E-mediated cell lysis. The latter stimulated both humoral and cell-mediated immune responses. Thus, chickens receiving the vaccine showed increased antigen-specific IgG levels and lymphocyte proliferation, which suggests that it can be a safe and effective inactivated vaccine option for preventing virulent *S*P infection [104].

As a conclusion, we should not only focus on viable vaccine candidates and turn a blind eye on developed vaccines that did not make it to the commercial phase but stagnated in the research phase. As such, we can cite the *aroA* mutant in 1993, which was proven to not be sufficiently invasive, and the 1998 *nuoG* mutant that was not attenuated enough and where even inoculation with 107 viable organisms produced no effect [89,90]. Thus, in the future, we should encourage the use if these specific genetically engineered strains rather than vague mutants as vaccines to increase safety and afford a better control of FT and PD [88].

## 6. Reversion to Virulence of *S*G9R and Differential Diagnosis between Wild-Type *S*G and *S*G9R

The *S*G9R vaccine strain has been administered subcutaneously for a long time to protect against FT caused by *S*G and the food poisoning risk posed by *S*E infection in egg-laying hens in several countries [105]. The potential reversion to virulence of *S*G9R in hens has therefore been a continuing source of concern, and post-genomic techniques are currently the most widely used method for systematically analyzing gene expression and function [106,107].

Several studies demonstrated that *S*G9R did not revert to virulence after serial passages in chickens, and its use in vaccination did not cause clinical signs or death in young hens, with the exception of a slight reduction in weight gain [85]. However, some studies have indicated that there is a chance of a resurgence of virulence, particularly given that the vaccine’s recipient has the same plasmid thought to be crucial to virulence and that a nonsense mutation in the LPS 1,2-glucosyltransferase (*rfa*J) gene turned the *S*G9R strain into a rough one [108]. Nevertheless, the fact that the attenuation may be attributed to just one-point mutation raises questions about pathogenic reversion [87].

Numerous *S*G9R-like rough strains have been found, and prior research has shown that the field-isolated *S*G from *S*G9R-vaccinated farms display the same DNA fingerprint as *S*G9R [109]. This theory was confirmed through the isolation of *S*G9R rough strains from cases of avian typhoid in hens that had received the *S*G9R vaccine [88,110]. For instance, Van Immerseel and coauthors demonstrated that isolates from an outbreak of avian typhoid in Belgium were nearly identical to the strain used in the vaccine based on the results of pulsed field gel electrophoresis (PFGE) and multiple locus variable tandem repeat (MLVA) analysis. Furthermore, sequencing revealed that the *S*G9R strain is nearly identical to another field strain, except for a few differences, most notably in the pyruvate dehydrogenase *ace*E gene and *rfa*J, which supports the idea that the field strain descended from *S*G9R [88].

Additionally, Kwon et al. (2011) compared between *S*G9R and wild strains of *S*G when it comes to genes responsible for LPS biosynthesis (*rfa*J and *rfa*Z) and those responsible for virulence (spv cluster and SPI-2). They revealed that *S*G9R included both a distinct *rfa*J nonsense mutation (TCA to TAA) and a shared rfaZ mutation (G-deletion) across the rough and smooth *S*. Gallinarum strains. Moreover, the presence of several intact or functioning virulence genes in *S*G9R’s chromosome, including *spv*B, *spv*C, and *inv*A, as well as the *Salmonella* pathogenicity islands SPI-1 and SPI-2, has also been shown to be correlated with the presence of residual virulence [108].

In order to identify genes associated with virulence changes, Kang and co-authors compared the proteome and transcriptome of *S*G9R to those of two wild-type strains. The proteome study revealed that *S*G9R is deficient in nine proteins, one of which is related to pathogenicity. The transcriptome study of *S*G9R identified 24 upregulated and 97 downregulated genes with 50% involved in virulence pathways. This study revealed that *S*G9R attenuation may be accompanied by a combination of defective virulence components; hence, reversion to virulence would not be the result of a single mutation event [111]. In a separate investigation, Kang et al. (2012 developed a triplex PCR technique to distinguish between *S*G, *S*P, and *S*G9R. Through the development of a suppression subtractive hybridization (SSH) library, the researchers identified sequences exclusive to *S*G9R that are absent or divergent in the wild *S*G strain. Suppression subtractive hybridization clones (718 clones), which were successfully sequenced, yielded a total of 565 non-redundant insertions. Sequences of 14 inserts were exclusive to the *S*G9R strain. However, SNPs discovered in another insert (9R22C9) were more advantageous for strain differentiation [112].

More recently, a study conducted by Beylefeld et al. (2023) using the SNP phylogenetic analysis revealed that nine strains associated with outbreaks of FT in South Africa were genetically closer to vaccine strains than wild-type *S*G9 strains, and four of them showed specific SNPs in the genes *ace*E and *rfa*J, which are markers of attenuation. These four isolates also retained intact *spv*, *SPI-1*, and *SPI-2* gene clusters, providing conclusive evidence that they were originally vaccine strains which reverted to virulence. On the other hand, five other field isolates lacked the *S*G9R attenuation markers. However, variant analysis identified certain genetic characteristics (an SNP in the *yihX* gene, insertions in the *ybj*X and *hyd*H genes, and deletions in the *fts*K and *sad*A genes) shared between these field isolates and vaccine strains, but not present in wild-type *S*G9. This finding indicates that these five field isolates were also likely revertant vaccines [113].

Thus, the *S*G9R vaccine in hens has raised concerns about its potential to regain virulence. Research on gene expression, proteomics, and transcriptomics suggests that attenuation involves multiple factors. However, ongoing vigilance and research are crucial to understand and manage the potential risks associated with *S*G9R’s virulence reversion [87].

## 7. Livestock Hygiene Practices as Measure of Control of *Salmonella* Infection in Poultry

Control strategies of SPG can be different from one country to another: the European Union focuses on production conditions and the environment while the United States focuses on processed products. Others combine the two types of actions [114]. Biosecurity measures applied at the farm level focus on several actions, notably the following: (i) installation of units in areas with a low concentration of livestock; (ii) establishment of rigorous sanitary barriers integrating aspects of design of facilities and equipment, protection, cleaning and disinfection with dedicated shoes and clothing for each room, hand washing protocols, separation of clean/dirty sectors, treatment food by heat, food analysis, and monitoring of staff and visitors; (iii) control of vermin: insects, rodents, birds, and other wild animals, (iv) poultry houses and equipment should be thoroughly cleaned and disinfected prior to use for a new lot of birds, (v) hatching eggs should be collected from the nests at frequent intervals to aid in the prevention of contamination with disease-causing organisms; and (vi) day-old chicks, poults, or other newly hatched poultry should be distributed in clean, new boxes and new chick papers. All crates and vehicles used for transporting birds should be cleaned and disinfected after each use [115].

## 8. Conclusions

*Salmonella* Gallinarum and *S*P are significant bacterial pathogens that pose a serious threat to animal health. The increasing prevalence of multidrug-resistant strains and the complex virulence mechanisms of *S*G and *S*P are challenges for control and treatment of these diseases. However, the development of newer and more sensitive diagnostic methods provides hope for better management and control of these pathogens.

Further research into the pathogenesis of *S*G and *S*P and the mechanisms underlying their drug resistance and virulence is crucial to developing effective prevention and treatment strategies. The integration of epidemiological surveillance, proper vaccination protocols and good management practices are essential to prevent and control the spread of these pathogens in poultry populations.

In conclusion, continued research and collaboration between researchers, veterinarians, and public health agencies are necessary to cope with the significant economic and eventual public health impacts of *S*G and *S*P. With a better understanding of its pathogenesis and the development of novel prevention and treatment strategies, mitigation of the threat posed by *S*G and *S*P can be achieved.

## Figures and Tables

**Table 1 antibiotics-13-00023-t001:** Summary of studies of AMR emergence.

Study	Country	Year	*S*G/*S*P	Number of Isolates	Phenotypical Antimicrobial Resistance
[38]	Korea	2014–2018	*S*G	130	AMP (28.6%), AUG2 (10.7%), NAL (100.0%), CIP (50.0%), CHL (17.9%), and COL (14.3%), CF (21%), FOX (15%), TET (11%), GEN (68%), SXT (14%).
[39]	Korea	2013–2018	*S*G	30	STR (100%), FIS (100%), COL (100%), NAL (96.7%), CIP (90%), GEN (66.7%), CHL (3.3%), AMP (3.3%), TET (3.3%), FFC (3.3%).
[41]	China	1962–2010	*S*P	337	AMP (34.4%), CAB (25.5%), CFM (46.6%), CTX (2.4%), STR (61.7%), GEN (5.3%), KA (3.9%), SPC (45.0%), CHL (4.1%), TET (58.7%), SMX (52.8%), TMP (82.8%), SXT (49.4%), NAL (69.0%), CIP (4.5%), NIT (26.4%).
[44]	China	1962–2007	*S*P	450	AMP (40.2%); CAB (39.1%); GEN (2.5%); KA (2.5%); STR (58%); CHL (1.8%); TET (58.9%); TMP (93.1%); SXT (24.2%); ENR (6.7%); CIP (0.4%); NAL (19.3%).
[46]	Iran	2012–2017	*S*G	60	PE (100%); AMX (75%); AUG2 (50%); NIT (80%); NAL (45%); CIP (37%); CHL (20%); NEO(30%); KA (0%); SXT (0%); COL (0%).
[45]	Brazil	2006–2013	*S*G	24	Intermediate to Resistant: AUG2 (0%), CTX (0%), IMP (0%), CAZ (0%), CFP (0%), ETP (0%), CEF (0%), TET (8% R), ETP (0%), FFC (4%), SXT (4%), NAL (42%), CIP (34.4%), ENR (33%).
	Brazil	2006–2013	*S*P	17	Intermediate to Resistant: AUG2 (0%), CTX (0%), ATM (0%), CAZ (0%), CFP (0%), ETP (0%), CEF (0%), TET (6%), ETP (0%), FFC (0%), SXT (0%), NAL (35%), CIP (29%), ENR (6%).

In studies where only susceptibility is given, both intermediate and resistant phenotypes were considered. AMP: ampicillin; AMX: amoxicillin; ATM: aztreonam, AUG2: amoxicillin/clavulanic acid; CAB: carbenicillin; CAZ: ceftazidime; CEF: ceftiofur; CF: cephalotin; CFP: cefepime; CHL: chloramphenicol; CIP: ciprofloxacin; CFM: cefamandol; COL: colistin; CTX: cefotaxime; ENR: enrofloxacin; ETP: ertapenem; FFC: florfenicol; FIS: sulfisoxazole; FOX: cephoxitin; GEN: gentamicin; KA: kanamycin; NAL: nalicixic acid; NIT: nitrofurantoine; NEO: neomycin; PE: penicillin; SMX: sulfamethoxazole; SPC: spectinomycin; STR: streptomycin; SXT: trimethoprim-sulfamethoxazole; TET: tetracycline; TMP: trimethoprim.

**Table 2 antibiotics-13-00023-t002:** List of primers used to detect different *S*G and *S*P genes/loci.

Targeted Gene/ Locus	Protein Encoded	Primer	Sequence (5′ to 3′)	Amplicon Size (bp)	PCR Conditions	*S*G	*S*P	Reference
*rhs* locus 2	Type II toxin-antitoxin	*rhs* (F)	TCGTTTACGGCATTACACAAGTA	402	95 °C for 5 min; 25 cycles at 95 °C for 30 s, 56 °C for 45 s, and 72 °C for 50 s; and 72 °C for 10 min	+ ^*^	+	[54]
		*rhs* (R)	CAAACCCAGAGCCAATCTTATCT					
*ste*B gene	*Fimbrial* ushers	*steB* (F)	TGTCGACTGGGACCCGCCCGCCCGC	636		+	−	
		*steB* (R)	CCATCTTGTAGCGCACCAT					
*stn* gene	Enterotoxin	*stn* (F)	TATTTTGCACCACAGCCAGC	131	94 °C for 5 min; 30 cycles of 94 °C for 45 s, 52 °C for 45 s, and 72 °C for 40 s; and 72 °C for 10 min	+	+	[62]
		*stn* (R)	CGACCGCGTTATCATCACTG					
*I137_08605*gene	Unknown	*I137_08605* (F)	CACTGGAGACTCTGAGGACA	290		+	+	
		*I137_08605* (R)	GGGCAGGGAGTCTTGAGATT					
*rat*A gene	RNA Antitoxin A	*ratA* (F)	ATTGCTCTCGTCCTGGGTAC	571		+	−	
		*ratA* (R)	TACCGATACGCCCAACTACC					
*cig*R gene	T3SS2 effector	*cigR* (F)	ATGAATAATCGTCGTGGTTT	421	95 °C for 3 min; 30 cycles of 95 °C for 15 s, 50 °C for 15 s, and 72 °C for 30 s, and 72 °C for 10 min	+	+	[53]
		*cigR* (R)	TAATAATCGCCGTGACCACC					
*ipa*J gene	T3SS effector	*ipaJ* (F)	TACCTGTCTGCTGCCGTGA	741	95 °C for 3 min; 30 cycles at 95 °C for 30 s, 58 °C for 45 s, and 72 °C for 50 s; and 72 °C for 10 min	−	+	[63]
		*ipaJ* (R)	ACCCTGCAAACCTGAAATC					
*glg*C gene	Glycogen biosynthesis	*glgC* (F)	TGGAGAGGATAATCCGGTGA	252	94 °C for 5 min; 30 cycles of 94 °C for 30 s, 55–65 °C for 30 s, and 30 s at 72 °C for 30 s; and 72 °C for 7 min	+	−	[61]
		*glgC* (R)	ATCAACACCATCCGCAATTT					
*spe*C gene	Ornithine decarboxylase	*speC* (F)	CCCGTTGCACATTAATCCTT	174	94 °C for 5 min; 30 cycles of 94 °C for 30 s, 55–65 °C for 30 s, and 72 °C for 30 s; and 72 °C for 7 min	+	+	
		*speC* (R)	CGGAGCTGGTATCCAGTTTG					
*rfb*S gene	paratose synthetase	*rfbSF* (F)	GTATGGTTATTAGACGTTGTT	187	94 °C for 5 min; 25–30 cycles of 94 °C for 1 min, 45 °C for 1 min, and 72 °C for 2 min; and 72 °C for 5 min.	+	−	[59]
		*rfbSG* (R)	TATTCACGAATTGATATACTC					
		*rfbSF* (F)	GTATGGTTATTAGACGTTGTT	187		−	+	
		*rfbSP* (R)	TATTCACGAATTGATATATCC					
*rfb*S gene		*rfbS-SP* (F)	GATCGAAAAAATAGTAGAATT	147	94 °C for 5 min, 30 cycles of 94 °C for 1 min, 62 °C for 1 min, 72 °C for 1 min; and 72 °C for 5 min.	−	+	[60]
		*rfbS-SP* (R)	GCATCAAGTGATGAGATAATC					

* (+): amplified; (−): not amplified.

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
