# Peer review of "Salmonella enterica Serovar Gallinarum Biovars Pullorum and Gallinarum in Poultry: Review of Pathogenesis, Antibiotic Resistance, Diagnosis and Control in the Genomic Era"

_antibiotics, 2023, doi:10.3390/antibiotics13010023_

Round 1

Reviewer 1 Report

Comments and Suggestions for Authors

This review is well-structured and well-written. 

Authors described characteristic, infection, diagnosis and control and drug-resistence of Salmonella enterica in poultry, a problematic infection in livestocks as also in public health.

Authors selected appropriate references to define the global situation. Differently from other published material, in this review there are several important section concerning Salmonella and its non-neglibile characteristic for diagnosis and control, such as: virulence, multi-drug resistence, new diagnosis approach, vaccination.

Authors could address a specific paragraph on livestock hygiene pratics as measure of control of Salmonella infection in poultry.

Author Response

Answer to the Reviewers

We thank the Reviewers for their valuable comments and the opportunity to improve our manuscript and resubmit to Antibiotics. We have carefully considered the comments and revised the manuscript accordingly. The following is a point by point response to the all the Reviewers’ questions and comments.

Reviewer 1

Comment 1: Authors described characteristic, infection, diagnosis and control and drug-resistence of Salmonella enterica in poultry, a problematic infection in livestocks as also in public health. Authors selected appropriate references to define the global situation. Differently from other published material, in this review there are several important section concerning Salmonella and its non-neglibile characteristic for diagnosis and control, such as: virulence, multi-drug resistence, new diagnosis approach, vaccination.

Authors could address a specific paragraph on livestock hygiene pratics as measure of control of Salmonella infection in poultry.

We would like to thank you for your thorough review of our manuscript. We Agree, we have, accordingly, added a paragraph related to the livestock hygiene pratics as measure of control of Salmonella infection in poultry in lines 609 to 624, it’s read as follow : “Control strategies of SPG can be different from one country to another: the European Union focuses on production conditions and the environment while the United States focuses on processed products. Others combine the two types of actions (Feuillet, 2007). Biosecurity measures applied at the farm level focus on several actions, notably: i) Installation of units in areas with a low concentration of livestock; ii) establishment of rigorous sanitary barriers integrating aspects of design of facilities and equipment, protection, cleaning and disinfection with dedicated shoes and clothing for each room, hand washing protocols, separation of clean/dirty sectors, treatment food by heat, food analysis, monitoring of staff and visitors; iii) Control of vermin: insects, rodents, birds and other wild animals, iv) Poultry houses and equipment should be thoroughly cleaned and disinfected prior to use for a new lot of birds, v) Hatching eggs should be collected from the nests at frequent intervals and, to aid in the prevention of contamination with disease-causing organisms and vi) Day-old chicks, poults, or other newly hatched poultry should be distributed in clean, new boxes and new chick papers. All crates and vehicles used for transporting birds should be cleaned and disinfected after each use (NPIP, 2017).’’

Reviewer 2 Report

Comments and Suggestions for Authors

The review “Salmonella enterica serovar Gallinarum biovars Pullorum and Gallinarum in poultry: review of diagnosis and control in the Genomic Era” is very comprehensive with tons of information. The topic is very interesting and covers different aspects including Salmonella Gallinarum/Pullorum’s genome and its relation with virulence, Multi-drug resistance, Diagnostic advances in Salmonella Gallinarum/Pullorum, Vaccination against FT and PD etc. The review may be a good reference material for researchers working on the field. The writing is also good. I do not have any major issues with the manuscript. Below are some comments in the spirit of helping the authors improve the manuscript.

 1.      The authors are requested to discuss about genetic mechanism of multi-drug resistance in Salmonella with some recent references and discussion on bacterial biofilms

 2.      In Diagnostic advances in Salmonella Gallinarum/Pullorum the authors mentioned multiple polymerase chain reaction (PCR), Pulsed-Field Gel Electrophoresis (PFGE), DNA microarray, and sequence-based techniques. The authors discussed Polymerase Chain Reaction and Next Generation Sequencing. The authors didn’t discuss about Pulsed-Field Gel Electrophoresis (PFGE), DNA microarray. The authors are requested to throw some light on those techniques with some recent references. Moreover, what are the limitations of each technique? Kindly mention.

 3. The authors are requested to mention future research areas related to the topic in a separate paragraph. Although the authors mentioned few points in conclusion section, it will be great if the authors can pinpoint few more areas where immediate research is required with reasonable discussion. 

Author Response

Answer to the Reviewers

We thank the Reviewers for their valuable comments and the opportunity to improve our manuscript and resubmit to Antibiotics. We have carefully considered the comments and revised the manuscript accordingly. The following is a point by point response to the all the Reviewers’ questions and comments.

Reviewer 2

Comment 1:   Discuss about genetic mechanism of multi-drug resistance in Salmonella with some recent references and discussion on bacterial biofilms.

We agree with you. In the new version, genetic mechanisms are discussed, both inherited and acquired in a dedicated paragraph with resistance mechanisms for each antibiotic family (Lines 127 to 158). It reads as follows: There are two types of antibiotic resistance: innate and intrinsic. The intrinsic mecha-nism of resistance includes the following mechanisms: change of the antibiotic target site, permeability of cell membranes, efflux pumps that carry antibiotic molecules out of the cell, and antibiotic inactivation [26]. Conversely, acquired resistance results from bacteria appropriating genetic material through horizontal gene transfer. Bacte-ria can acquire genes that confer antibiotic resistance to develop an antibiotic-resistant phenotype. These genes are carried and transferred by integrons, transposons, plas-mids, and prophages [27], [28]. Different resistance mechanisms are deployed by Sal-monella depending on the antibiotic family in question [28] . For instance, resistance against aminoglycosides, which acts by binding on the ribosome, can be provided by: (i) methylation of the subunit 30S of the ribosome, (ii) aminoglycoside acetyltransfer-ase coded by aac(6′)-Ib, (iii) decreased permeability [29], [30]. Resistance against β-lactams, which acts by interfering with the synthesis of peptidoglycan, is provided by: (i) enzymatic inactivation through the hydrolysis of β-lactamase, (ii) expression of the β-lactams resistance gene bla, (iii) β-lactamase point mutation, which produces an extend-ed-spectrum β-lactamase (iv) Overexpression of the efflux pump genes macAB, mdtABC, emrAB, mdtK, and acrD, (v) Diminished permeability [31]. Quinolones act by interfering with bacterial DNA replication and transcription, and resistance against them is provided by: (i) expression of the quinolone resistance genes parC and gyrA, (ii) reduced activity by the aac(6′)-lb-cr gene expression, (iii) efflux pumps encoded by the oqxAB and qepA genes [32]. Macrolides and chloramphenicol bind to the 50S subunit of ribosomes and inhibit the production of proteins. Mechanisms responsible for the re-sistance against macrolides are: (i) enzymatic inactivation by phosphotransferase or esterase, (ii) mutations in the 23S rRNA gene, (iii) efflux pumps genes mef and msr [33], [34]. Resistance against chloramphenicol is provided by: (i) expression of the floR re-sistance gene, (ii) enzymatic inactivation, (iii) overexpression of acrAB-tolC efflux sys-tem[34], [35]. Resistance mechanisms against tetracyclines (TET), which bind to the 30S ribo-somal subunit and inhibit protein synthesis, are: (i) target site mutation in ri-bosomes, (ii) expression of the genes tetA, tetG, and tetB, (iii) efflux pump against TET [35]. Sulfonamides prevent bacteria from producing the B vitamin folate, and the re-sistance mechanism against them is the expression of sul1, sul2 and sul3 genes encoding dihydropteroate synthetase (DHPS) with low affinity for sulfonamides [36].

For bacterial biofilm, it’s also discussed with a study that proves the positive correlation between biofilm forming and antimicrobial resistance (lines 235 to 244). It reads as follows: “This study also highlighted the correlation between antibiotic resistance and biofilm forming.

The latter is a set of an extracellular matrix and persistent cells that we can find both inside and outside the host body [47]. The capacity of bacteria to form biofilms affords a favorable exchanging space where the frequency of genetic material ex-change is more important. Thus, genes encoding resistance are transferred among bio-film-forming bacteria with MDR traits [48,49]. This was confirmed by the results of the study by Farahani et al. (2023) where the findings demonstrated a positive correla-tion between the degree of biofilm formation and certain resistance genes, including Fox M, GES, Fox, KPC, and qnrB [46].’’

Comment 2: The authors didn’t discuss about Pulsed-Field Gel Electrophoresis (PFGE), DNA microarray. The authors are requested to throw some light on those techniques with some recent references. Moreover, what are the limitations of each technique?

We thank the reviewer for your kind comments. The Reviewer is correct. We added a paragraph including the limitations of these techniques (Lines 267 to 272). It’s read as follow: “Genotypic techniques accessing genetic material from chromosomal and extrachromosomal DNA allow for differentiation between closely related strains. These include multiple polymerase chain reaction (PCR), Pulsed-Field Gel Electrophoresis (PFGE), DNA microarray, and Whole Genome Sequencing (WGS) [51]. Basically, in epidemiological research, PCR and WGS are the main used techniques due to limitations of the other ones like PFGE which demonstrates only minor differences between the strains [52]”.

Comment 3: future research areas related to the topic in a separate paragraph. Although the authors mentioned few points in conclusion section, it will be great if the authors can pinpoint few more areas where immediate research is required with reasonable discussion.

We agree with the Reviewer. The future research areas are mainly concerning vaccination, which was already mentioned in the vaccination part as now presented in lines 534 to 541, it’s read as follow: “As a conclusion, we should not only focus on viable vaccine candidates and shed a blind eye on developed vaccine that didn’t make it to the commercial phase and stag-nated in the research phase. As such, we can cite the aroA mutant in 1993, that was proved not sufficiently invasive, and the 1998 nuoG mutant that wasn’t enough atten-uated and where even inoculation with 107 viable organisms produced no effect [89,90]. Thus, in the future, it should be encouraged to use these specific genetically engineered strains rather than vague mutants as vaccines to increase safety and afford a better control of FT and PD [88].’’

Reviewer 3 Report

Comments and Suggestions for Authors

Please revise the title as the manuscript covers not only the diagnosis and control but also the pathogenesis.

When introducing FBD for the first time, provide its full name.

All gene names need to be in italics. Please check throughout the manuscript.

66-117. Consider presenting the information in a tabular format for a clearer representation of the emergence or trends in antimicrobial resistance.

For Table 1, include an additional column specifying the protein encoded by each gene and another column for the PCR conditions.

Comments on the Quality of English Language

The overall quality of English is good however, there are some errors that need to be addressed, such as those found in lines 22-23, 51-52, and other errors throughout the manuscript.

Author Response

Answer to the Reviewers

We thank the Reviewers for their valuable comments and the opportunity to improve our manuscript and resubmit to Antibiotics. We have carefully considered the comments and revised the manuscript accordingly. The following is a point by point response to the all the Reviewers’ questions and comments.

Reviewer 3

Comment 1: revise the title as the manuscript covers not only the diagnosis and control but also the pathogenesis.

We agree with the reviewer, we have now modified the title, it’s read as follow: “Salmonella enterica serovar Gallinarum biovars Pullorum and Gallinarum in poultry: review of pathogenesis, antibiotic resistance, diagnosis and control in the Genomic Era’’

Comment 2: All gene names need to be in italics

Thank you for pointing this out. We have corrected it now in the manuscript.

Comment 3: When introducing FBD for the first time, provide its full name.

We agree with the Reviewer and now we have corrected it. Please see the manuscript. (Line 42)

Comment 4: 66-117. Consider presenting the information in a tabular format for a clearer representation of the emergence or trends in antimicrobial resistance.

We agree with the Reviewer and we have now clarified this important point in the text. A table summarizing the studies is well documented in the new version (lines 249-250).

Comment 5: For Table 1, include an additional column specifying the protein encoded by each gene and another column for the PCR conditions.

We agree with the Reviewer and thank you for bringing this to our attention. The columns specifying the protein encoded by each gene and the PCR conditions are now added in table 2 in the manuscript.

Comment 6: The overall quality of English is good however; there are some errors that need to be addressed, such as those found in lines 22-23, 51-52, and other errors throughout the manuscript.

All mistakes through the document were corrected now;